

# A generalized spatial autoregressive neural network (GSARNN) method for three-dimensional spatial interpolation

Junda Zhan[1#], Sensen Wu[1,2#], Jin Qi[1,2], Jindi Zeng[1], Mengjiao Qin[1,2], Yuanyuan Wang[3,2], Zhenhong Du[1,2]*

[1]School of Earth Sciences, Zhejiang University, Hangzhou 310027, China;

[2]Zhejiang Provincial Key Laboratory of Geographic Information Science, Hangzhou 310028, China;

[3]Ocean Academy, Zhejiang University, Zhoushan 316022, China

*# These authors contributed equally to this work.*

*\* Correspondence to:* Zhenhong Du (duzhenhong@zju.edu.cn)

E-mail address of each author:

Junda Zhan: 11838010@zju.edu.cn

Sensen Wu: wusensengis@zju.edu.cn

Jin Qi: qijinjesse@zju.edu.cn

Jindi Zeng: 21738033@zju.edu.cn

Mengjiao Qin: qinmengjiao@zju.edu.cn

Yuanyuan Wang: wangyuanyuanxy@zju.edu.cn

Zhenhong Du: duzhenhong@zju.edu.cn





**Abstract.** Spatial interpolation, which is one of the most important spatial analysis methods, predicts unsampled spatial data from the values of sampled points. Generally, the core of spatial interpolation is fitting spatial weights via spatial correlation. Traditional methods express spatial distances in a conventional Euclidean way and conduct relatively simple spatial weight calculation processes, limiting their ability to fit complex spatial nonlinear characteristics in multidimensional space. To tackle these problems, we developed a generalized spatial distance neural network (GSDNN) unit to generally and adaptively express spatial distances in complex feature space. By combining the spatial autoregressive neural network (SARNN) with the GSDNN unit, we constructed a generalized spatial autoregressive neural network (GSARNN) to perform spatial interpolation in three-dimensional space. The GSARNN model was examined and compared using two three-dimensional cases: a simulated case and a real Argo case. The experiment results demonstrated that exploiting the feature extraction ability of neural networks, the GSARNN achieved superior interpolation performance and was more adaptable than inverse distance weighted, ordinary Kriging, and SARNN methods.

## 1. Introduction

Due to the difficulties of establishing abundant observation stations and the existence of unobservable positions in space, research areas in geospatial subjects typically contain many unsampled data points. Estimating unknown data based on sampled point values and expanding discrete and sparse data into continuous data are the main goals of spatial interpolation models. Spatial interpolation is widely applied in many research fields, including air quality (Tang et al., 2017), climate (Arowolo et al., 2017; Adhikary et al., 2017), hydrology (Cheng et al., 2017), marine environment (Gao et al., 2020; Zhang et al., 2021), ecosystem (Pan et al., 2021), etc. in the field of natural environment and land/house price (Hu et al., 2013; Szczepańska et al., 2020), traffic (Ma et al., 2019), urban noise (Aumond et al., 2018), agriculture (da Silva Júnior et al., 2019), etc. in the field of social development. Therefore, accurately fitting the spatial correlation between elements and improving model spatial interpolation abilities are important for exploring spatial distribution patterns and change trends and solving myriad problems encountered in nature and society.

Tobler's First Law of Geography (Tobler, 1970) proposes the existence of spatial correlation, which is a general feature of geospatial space as well as a core theory supporting spatial interpolation modeling. In spatial interpolation, generally, the closer objects are in space, the more likely it is that their features will be similar. Therefore, the precision of spatial element correlation weights is the key to spatial interpolation modeling and determines the reliability of interpolation prediction, while spatial distance expression is the basis of spatial weight fitting. In fact, interpolation can be regarded as the problem of mining complicated nonlinear relationships between spatial distances and spatial weights.

There are various traditional spatial interpolation methods, such as inverse distance weighted (IDW), Kriging, natural neighbor, spline, trend surface and radial basis function, which are applied in particular scenarios. They can fit the relationships between spatial distance and spatial weight to a certain extent. However, these methods are based on simple mathematical formulas and parameter calculations and have difficulty describing nonlinear and complex relationships in spatial processes. These limitations prevent traditional interpolation approaches from accurately reflecting the relevant characteristics of geographical elements, restricting their spatial interpolation abilities. Many studies have improved and reformed the traditional methods based on their principles. Li et al. (2020) incorporated data-to-data distance into IDW to account for the spatial configuration of neighborhoods and proposed the dual IDW (DIDW) method. Kumar et al. (2012) applied geographically weighted



regression theories to Kriging and applied the proposed GWRK method to the spatial modeling of soil organic carbon stock. Li (2018) used particle swarm optimization (PSO) algorithm to fit the mutation function and introduced top-k query to realized global optimization of parameters based on ordinary Kriging, developing the PT-Kriging method. In recent years, machine learning and neural network theories develop rapidly, which provide new solutions for accurate spatial interpolation. A number of theories and strategies of machine learning were introduced to solve interpolation problem, such as random forest (da Silva Júnior et al., 2019; Sekulić et al., 2020), support vector machine (Li et al., 2018) and neural network (Sivapragasam et al., 2010). Zeng et al. (2020) proposed the spatial autoregressive neural network (SARNN) model for two-dimensional spatial interpolation by integrating the neural network with spatial autoregression, achieving superior performance compared with traditional spatial interpolation methods.

With regard to spatial distance expression, traditional methods and SARNN model employ Euclidean distances calculated using a fixed formula, treating all directions in space equivalently. However, spatial anisotropy, the universal feature of spatial element distribution and change, should be considered for accurate spatial interpolation, especially in three-dimensional space (Wu et al., 2020). For example, mineral resource distribution exhibits directional differences affected by geological structures (Samal et al., 2011), soil nutrient content gradients have specific orientation patterns (Abd El-Hady et al., 2018), and climate elements such as surface temperature and precipitation can be strongly direction-dependent on spatial scales (Chen et al., 2016; Zhang et al., 2017; Wang et al., 2018). In three-dimensional spatial interpolation, spatial isotropic distance expression implies that any point with the same distance from a target point will exert the same effect on it, even if they are from different directions. It ignores the effects of differences and the complex coupling of various spatial axes on spatial weights, resulting in insufficient spatial correlation mining.

To address these limitations, we propose a generalized spatial distance neural network (GSDNN) unit to express distances in multidimensional space with nonlinear characteristics. In the GSDNN, generalized spatial distances between elements are fitted using multidirectional distance components. Furthermore, by combining the GSDNN unit with the SARNN to integrate generalized distances into the spatial interpolation method, we developed a generalized spatial autoregressive neural network (GSARNN) model to realize complex nonlinear spatial interpolation modeling in three-dimensional space, improving spatial interpolation prediction and fitting abilities.

The remaining sections of this paper are organized as follows. Section 2 briefly introduces two traditional interpolation methods, defines the SARNN model and GSDNN unit, and describes the overall GSARNN model framework, training strategy, and evaluation method. In Section 3, we perform interpolation experiments on two cases and compare the IDW, Kriging, SARNN, and GSARNN model results. The discussion and conclusion are given in Section 4 and Section 5, respectively.

## 2. Generalized spatial autoregressive neural network

### 2.1 Traditional spatial interpolation

Interpolation methods can be divided into deterministic interpolation and geostatistical interpolation approaches, according to their mathematical principles. Deterministic interpolation, such as IDW, spline, and trend surface methods, builds the fitting surface according to the smoothness of the whole spatial surface or the similarities of spatial information elements to predict data in unknown regions. Geostatistical interpolation, such as the Kriging method, builds the sample point spatial structure by analyzing the distribution laws and relevant features of the sample points in space and predicting the change trend of the whole spatial area.





### 2.1.1 IDW interpolation

IDW interpolation (Shepard, 1968) is a deterministic interpolation method (Watson and Philip, 1985). IDW regards the value at an unsampled location as the distance-weighted average of the sampled point values (Longley et al., 2011). For an unsampled point, the closer the sampled point is, the greater an influence it exerts; the influence is inversely proportional to the distance 5 (Tan and Xu, 2014). IDW can be expressed as:

$$\hat{z}_i = \sum_{j=1}^{n} \frac{\frac{1}{(D_{ij})^P}}{\sum_{j=1}^{n} \frac{1}{(D_{ij})^P}} z_j, \, i = 1,2,\cdots,m \tag{1}$$

where $\hat{z}_i$ is the predicted value at the unsampled point $i$, $z_j$ is the observed value of point $j$, $D_{ij}$ is the Euclidean distance between point $i$ and point $j$, and $P$ is the power parameter that defines the weight decline rate as the distance increases. By defining a larger $P$, the influence of closer points is strengthened, affecting the smoothness of the interpolation results.

Due to the simplicity, convenience, and intuitiveness of the IDW method, it has been widely used in many fields, including 10 geography, agriculture, oceanography, and environmental studies; however, extreme values among the sampled points can have a substantial impact on IDW spatial prediction results.

### 2.1.2 Kriging interpolation

Kriging methods, such as ordinary Kriging (OK), universal Kriging, and co-Kriging, are spatial interpolation methods designed to solve the problems of deposit reserves and error estimation (Krige, 1952; Matheron, 1963). These methods generate 15 unbiased optimal variable estimations in a finite area using the variation function to perform moving average interpolation according to the differences of the sample points' positions and spatial correlation degree. Among the Kriging methods, OK is the most commonly used.

Kriging can be expressed as:

$$z^*(x_0) = \sum_{i=1}^{n} \lambda_i z(x_i) \tag{2}$$

where $z^*(x_0)$ is the predicted value, and $\lambda_i$ and $z(x_i)$ are, respectively, the weight coefficient and observed value of point $i$.

Kriging methods involve the calculation of the weight coefficient $\lambda_i$, for which the key is to satisfy the unbiasedness and optimality. Unbiasedness means that $z^*(x_0)$ is the unbiased estimate of $z(x_i)$, that is:

$$E[z^*(x_0)] = E[z(x)] \tag{3}$$

Optimality means that $z^*(x_0)$ is the optimal estimate of $z(x_i)$, and the variance between the predicted value of the unsampled points and the estimated value of the observed points is the smallest, that is:

$$\min_{\lambda_j} Var(z^*(x_0) - z(x)) \tag{4}$$

### 2.2 SARNN model

Summarizing the principles of most traditional interpolation methods, it can be found that they are modeled following the core concept of "fitting the relationship between spatial distance and spatial weight", a relationship that is often complicated, containing nonlinear characteristics. Thus, achieving accurate fitting using only simple mathematical functions is difficult. Establishing a nonlinear expression between the spatial distance $D$ and the weight coefficient $w_{ij}$ is necessary to interpolate





unsampled points from observed points. The spatial weight of sampled points to point $i$ is defined as:

$$w_i = (w_{i1}, w_{i2}, \cdots, w_{in}) = f(d_{i1}^s, d_{i2}^s, \cdots, d_{in}^s) \tag{5}$$

where $w_i$ represents the spatial weight vector of point $i$, $w_{ij}$ is the spatial weight of point $j$ to point $i$, and $d_{ij}^s$ is the distance between point $i$ and point $j$.

To characterize complex nonlinear relationships in space, Zeng et al. (2020) designed the SARNN model, exploiting the powerful modeling and nonlinear fitting capabilities of neural networks to fit the spatial weight $w_i$.

It should be noted that since the elements on the diagonal of the weight matrix are the weights of the points to themselves, these weights should be set to 0 to avoid overfitting:

$$w_{ij} = \begin{cases} f(d_{i1}^s, d_{i2}^s, \cdots, d_{in}^s)_j, & if\ i \neq j \\ 0, & if\ i = j \end{cases} \tag{6}$$

To reset the weight on the diagonal, $w_{ij}$ is defined as:

$$w_{ij} = \rho_{ij} \times k_{ij} \tag{7}$$

where $\rho_{ij}$ is the spatial weight component, and $k_{ij}$ is the standard weight component, which ensures that the neural network weight is independent of the point itself in the training process. $k_{ij}$ can be expressed as:

$$k_{ij} = \begin{cases} 1, & if\ i \neq j \\ 0, & if\ i = j \end{cases} \tag{8}$$

Next, the problem of solving the spatial weight can be transformed into the problem of constructing and training the neural network. The distance from the point to be interpolated to the observed point is the network input, the hidden layers are defined, and the spatial weight vector $\rho_i$ is the output, that is:

$$\rho_i = (\rho_{i1}, \rho_{i2}, \cdots, \rho_{in}) = SARNN([d_{i1}^s, d_{i2}^s, \cdots, d_{in}^s]^T) \tag{9}$$

where $[d_{i1}^s, d_{i2}^s, \cdots, d_{in}^s]^T$ represents the vector of distances from point $i$ to other sample points, and $\rho_{ij}$ represents the spatial weight of point $j$ to point $i$. $\rho_{ij}$ is correspondingly multiplied by $k_{ij}$ to obtain the weight coefficient $w_{ij}$. The matrix form is as follows:

$$W = \rho * k = \begin{bmatrix} \rho_{11} & \rho_{12} & \cdots & \rho_{1n} \\ \rho_{21} & \rho_{22} & \cdots & \rho_{2n} \\ \vdots & \vdots & \ddots & \vdots \\ \rho_{n1} & \rho_{n2} & \cdots & \rho_{nn} \end{bmatrix} * \begin{bmatrix} 0 & k_{12} & \cdots & k_{1n} \\ k_{21} & 0 & \cdots & k_{2n} \\ \vdots & \vdots & \ddots & \vdots \\ k_{n1} & k_{n2} & \cdots & 0 \end{bmatrix} = \begin{bmatrix} 0 & \rho_{12}k_{12} & \cdots & \rho_{1n}k_{1n} \\ \rho_{21}k_{21} & 0 & \cdots & \rho_{2n}k_{2n} \\ \vdots & \vdots & \ddots & \vdots \\ \rho_{n1}k_{n1} & \rho_{n2}k_{n2} & \cdots & 0 \end{bmatrix} \tag{10}$$

The product of the final spatial weight matrix $W$ and the sampled value vector $y$ is the unsampled point estimation results. $\hat{y}$ can be expressed as:

$$\hat{y} = \begin{bmatrix} 0 & \rho_{12}k_{12} & \cdots & \rho_{1n}k_{1n} \\ \rho_{21}k_{21} & 0 & \cdots & \rho_{2n}k_{2n} \\ \vdots & \vdots & \ddots & \vdots \\ \rho_{n1}k_{n1} & \rho_{n2}k_{n2} & \cdots & 0 \end{bmatrix} \begin{bmatrix} y_1 \\ y_2 \\ \vdots \\ y_n \end{bmatrix} = Wy \tag{11}$$

### 2.3 GSARNN model

#### 2.3.1 Model definition

Spatial distance is the most important indicator of the relationship between two objects as well as the basis of spatial weight fitting. The essence of spatial interpolation is establishing a distance-based mapping relationship between the sampled region and the unsampled region.



For any two vectors $\alpha, \beta$ in the n-dimensional linear space $V$, there are a pair of coordinates $\alpha = (x_1, x_2, \cdots x_n)^T$ and $\beta = (y_1, y_2, \cdots y_n)^T$ under the orthonormal basis. There are many ways to define spatial distances, such as the Manhattan distance, Euclidean distance, and Minkowski distance, which can be expressed as:

$$D_{Manhattan} = |x_1 - y_1| + |x_2 - y_2| + \cdots + |x_n - y_n| \tag{12}$$

$$D_{Euclidean} = \sqrt{(x_1 - y_1)^2 + (x_2 - y_2)^2 + \cdots + (x_n - y_n)^2} \tag{13}$$

$$D_{Minkowski} = \sqrt[k]{(x_1 - y_1)^k + (x_2 - y_2)^k + \cdots + (x_n - y_n)^k} \tag{14}$$

The traditional two-dimensional spatial interpolation methods always use Euclidean spatial distance as the basis for expressing spatial correlation, treating different spatial relative positions equivalently. However, in geographic space—especially in three-dimensional and higher-dimensional spaces—the changing trend and speed of elements often differ along various axes, and there is local variability in the data. Using Euclidean distance for three-dimensional spatial interpolation is an isotropic solution (Allard et al., 2016) that reduces the dimensionality of the raw data, discards a large amount of relative position information between points, and cannot adequately reflect the complicated nonlinear characteristics of data change, restricting the accuracy

of interpolation in multidimensional linear space.

To solve these problems, we propose a generalized expression of spatial distance. The generalized spatial distance $d_{ij}^s$ of $\alpha = (x_1, x_2, \cdots x_n)^T$ and $\beta = (y_1, y_2, \cdots y_n)^T$ in n-dimensional linear space is defined as the function of the coordinate difference under the orthonormal basis, which can be expressed as:

$$d_{ij}^s = F(x_1 - y_1, x_2 - y_2, \cdots, x_n - y_n) = F(\Delta_1, \Delta_2, \cdots \Delta_n) \tag{15}$$

The distance components of the point to be interpolated $(x, y, z)$ and the known sample point $(x_i, y_i, z_i)$ under the three-

dimensional orthonormal basis are:

$$(dx, dy, dz) = (x - x_i, y - y_i, z - z_i) \tag{16}$$

To fully and adaptively capture the nonlinear effect of the elements' changing trend in three-dimensional space, we designed a GSDNN unit that generates generalized spatial distances considering anisotropy based on the distance components of each axis. It can be simply expressed as:

$$d_{ij}^G = F(\Delta_x, \Delta_y, \Delta_z) = GSDNN(dx, dy, dz) \tag{17}$$

Through network training, the generalized spatial distance automatically output by this network unit will reflect the complex

characteristics of the specific spatial elements. The GSDNN structure is shown in Fig. 1.

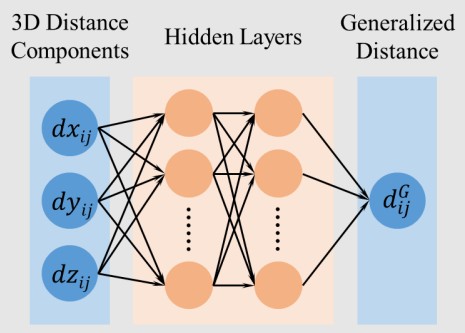

Figure 1. The GSDNN unit structure.

By replacing the input of the SARNN model with the GSDNN unit, Formula 11 can be refined as:





$$\rho_i = (\rho_{i1}, \rho_{i2}, \cdots, \rho_{in}) = GSARNN([d_{i1}^s, d_{i2}^s, \cdots, d_{in}^s]^T)$$
$$= SARNN([GSDNN(dx_1, dy_1, dz_1), \cdots, GSDNN(dx_n, dy_n, dz_n)]^T) \tag{18}$$

The refined model is the GSARNN model, and the overall model structure is shown in Fig. 2.

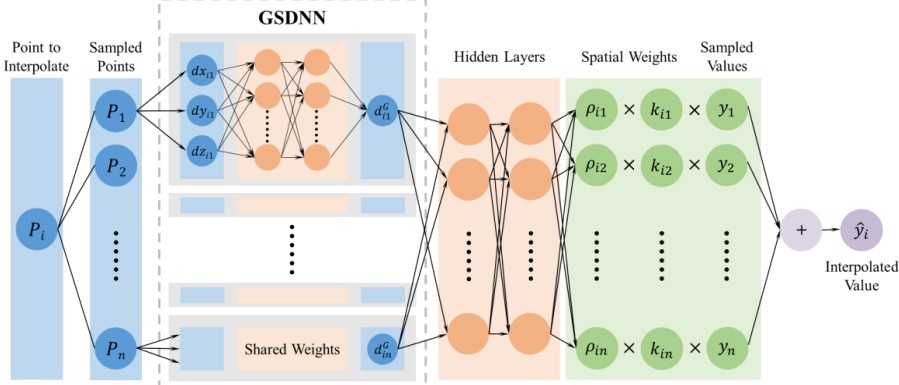

Figure 2. The GSARNN model structure.

In the modeling process, the distance components from the unknown point to the sampled points in three-dimensional space
are input into the GSDNN, and all GSDNN units share network weights and biases. Through the training process, the
generalized spatial distance between the two points under the specific spatial context of the interpolation element is output and
simultaneously becomes the input of GSARNN. After the hidden layer calculations, the output layer finally outputs the spatial
weight component, which is multiplied by the standard weight component and the observed values of the sampled points. The
sum of the output tensor is the interpolated value of the unsampled point. Note that the GSDNN unit separated from GSARNN
cannot be trained alone. In other words, the generalized spatial distance in the model has a specific meaning solely based on
the context of the specific spatial elements.

### 2.3.2 Model design and estimation

To improve the transferability and adaptability of the GSARNN and solve the problems of over-fitting and gradient vanishing
in neural network training, we design a set of model training strategy based on the cross-validation method, including the
overall training framework, parameter initialization method, activation function definition, and training optimization
algorithms. A complete set of training processes is established to improve training quality and interpolation accuracy, as shown
in Fig. 3.





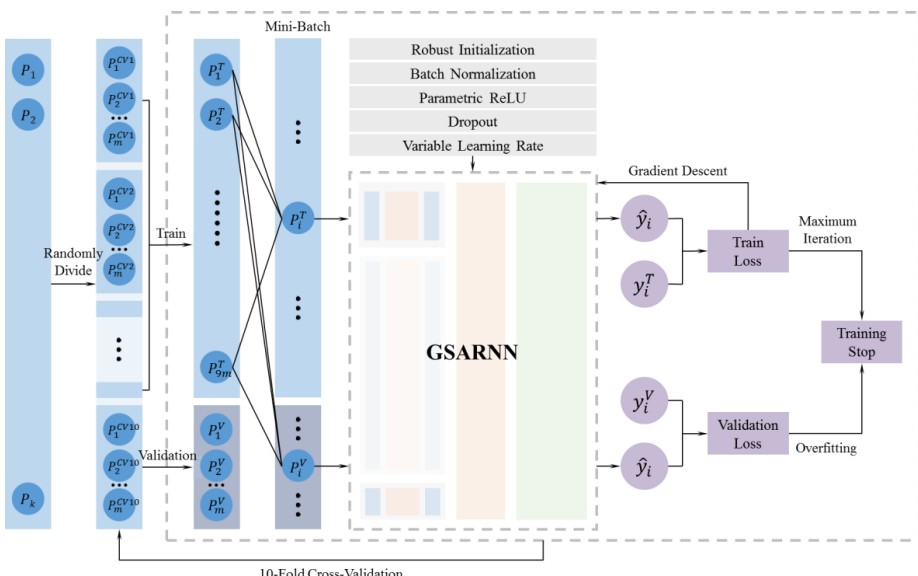

Figure 3. The network training framework of the GSARNN model.

We employ several neural network structure design and model optimization techniques to improve training efficiency. For each hidden layer, we first use the robust parameter initialization method proposed by He et al. (2015). Second, the batch
normalization method of Ioffe and Szegedy (2015) is adopted to accelerate the model training convergence speed and improve the training process stability. Third, the PReLU (Parametric Rectified Linear Unit) proposed by He et al. (2015) is used as the activation function to improve model performance. Finally, the dropout strategy developed by Srivastava et al. (2014) is integrated to strengthen the generalizability of the model.

### 2.3.3 Model training and validation

We use the 10-fold cross-validation method for model training. The dataset is randomly divided into 10 equal portions, among which nine portions serve as the training set, and the remaining portion is used as the validation set in turn. The training set is used to fit the data characteristics, and the validation set is used to verify the generalization performance of the model. The cross-validation method averages the training results of each group, reduces the sensitivity to data division, avoids overfitting to a certain extent, and extracts more effective features from the data.

Learning rate selection is critical in network training. An excessive learning rate will lead to an oscillation of the loss and unavailability of the optimal solution. Conversely, an insufficient learning rate will result in slow convergence or even gradient vanishing. In view of the characteristics of the GSARNN model, we adopt a custom variable learning rate in the training process. The formula is as follows:

$$\alpha = \begin{cases} \alpha_{start} + k_1 * epoch, & epoch < epoch_{up} \\ \alpha_{max}, & epoch \in \left[ epoch_{up}, epoch_{down} \right] \\ k_2^{epoch} * \alpha_{max}, & epoch > epoch_{down} \end{cases} \qquad (19)$$

where $\alpha_{start}$ is the initial learning rate, which increases gradually at the rate of $k_1$ until $\alpha_{max}$, and the maximum learning rate
is maintained for $n$ epochs. The learning rate then gradually decreases exponentially at the rate of $k_2$. The change of the learning rate throughout the training process is shown in Fig. 4.

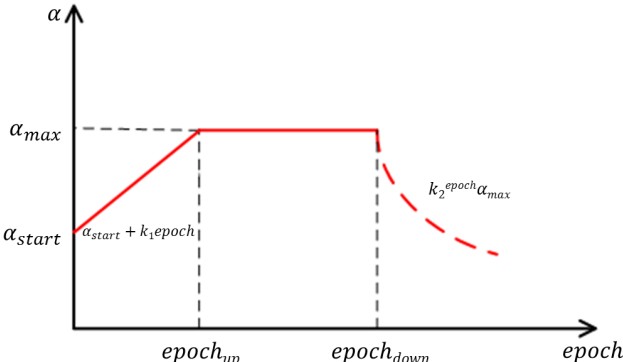

Figure 4. The variable learning rate change line.

The GSARNN model takes the mean square error (MSE) as the loss function in the training process:

$$loss = MSE = \frac{(\hat{y} - y)^2}{n} \tag{20}$$

### 2.4 Evaluation method

5     To quantitatively measure the performance of the IDW, OK, SARNN model, and GSARNN model methods, we use the determination coefficient ($R^2$), root mean square error (RMSE), mean absolute error (MAE), and mean absolute percentage error (MAPE) as evaluation indicators. The formulas are as follows:

$$R^2 = 1 - \frac{\sum_{i=1}^{n}(y_i - \hat{y}_i)^2}{\sum_{i=1}^{n}(y_i - \bar{y}_i)^2} \tag{21}$$

$$RMSE = \sqrt{\frac{\sum_{i=1}^{n}(y_i - \hat{y}_i)^2}{n}} \tag{22}$$

$$MAE = \frac{\sum_{i=1}^{n}|y_i - \hat{y}_i|}{n} \tag{23}$$

$$MAPE = \frac{1}{n}\sum_{i=1}^{n}\left|\frac{y_i - \hat{y}_i}{y_i}\right| \times 100\% \tag{24}$$

$R^2$ is a relative indicator that compares the model with the baseline using the average value as the interpolation result. The RMSE, MAE, and MAPE are absolute indicators that reflect the interpolation error; smaller values indicate higher model

10     accuracy.

### 3. Experiments and results

We use two three-dimensional datasets with distinct characteristics to test the interpolation performance of the GSARNN model in different scenarios, comparing it with the traditional IDW and OK methods and the SARNN model. In case one, we conduct experiments using a simulated dataset, which can be generated arbitrarily and controllably. By simulating a dataset

15     with complex features and conducting a quantitative cross-validation interpolation experiment on it, the feature extraction and fitting ability of the GSARNN model are fully and persuasively tested. In case two, we experiment on a measured Argo temperature dataset in the western Pacific area, which reflects the most authentic natural characteristics. In this case, in addition to the cross-validation interpolation, we select several spatial sections for interpolation prediction. By qualitatively analyzing the section interpolation results, the GSARNN model's ability to restore spatial element field patterns in practical interpolation

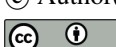



applications is examined.

### 3.1 Case one: simulated dataset

To examine the ability of the GSARNN model to handle data with complex characteristics in three-dimensional space, we combine trends of gradual change and sudden change to simulate a dataset in the three-dimensional spatial field, repeating the simulation and interpolation for 100 times.

### 3.1.1 Dataset

Defining the three-dimensional area of a cube with the unit length, the side length of the cube is 6, and the distance between adjacent data points is defined as 0.5. Therefore, the three-dimensional spatial research area contains $12 * 12 * 12 = 1728$ data points, and the spatial coordinates $(x_i, y_i, z_i)$ of each point are generated according to the following formulas:

$$x_i = \frac{1}{2} mod(i/12) \tag{25}$$

$$y_i = \frac{1}{2} int(mod(i/(12 * 12))/12) \tag{26}$$

$$z_i = \frac{1}{2} int(i/(12 * 12)) \tag{27}$$

The simulated value $V$ is defined as:

$$V = V_1 + \frac{V_2}{3} + \varepsilon \tag{28}$$

$V_1$ is a three-dimensional spatial data item with a gradual change trend, calculated as follows:

$$V_1 = 1 + \frac{1}{2}(x_i + y_i + z_i) \tag{29}$$

where $(x_i, y_i, z_i)$ are the coordinates of the sample point $i$ in the three-dimensional spatial field. $V_1$ has a gradual gradient along the $z = y = x$ direction, as shown in Fig. 5(a).

$V_2$ is a three-dimensional spatial data item with local spatial variability, calculated as follows:

$$V_2 = \begin{cases} 1 - \frac{1}{36}(9 - (3 - x_i)^2)(9 - (3 - y_i)^2)(9 - (3 - z_i)^2), \\ \qquad if \; dist[(x_i, y_i, z_i), (3,3,3)] \in [1.5,3] \\ 1 + \frac{1}{36}(9 - (3 - x_i)^2)(9 - (3 - y_i)^2)(9 - (3 - z_i)^2), else \end{cases} \tag{30}$$

$V_2$ has a spatial mutation at the two spherical surfaces 1.5 and 3 units away from the center of the three-dimensional space. The values of the entire dataset appear to diffuse from the center of the sphere to the surroundings, as shown in Fig. 5(b). $V_2$ possesses high local spatial variability, which adds certain challenges to the interpolation work.

$\varepsilon$ is a random item that brings some uncertainty to the simulated dataset and can be expressed as:

$$\varepsilon \sim \frac{1}{2} N(0,1) \tag{31}$$

The final three-dimensional spatial simulated dataset $V$ is generated by adding $V_1$, $V_2$, and $\varepsilon$, as shown in Fig. 5(c). The figure shows that the simulated data have both the gradient feature from $V_1$ and the mutation feature from $V_2$. By simulating such a dataset with specific data change characteristics and conducting interpolation experiments on it, this case compares the interpolation abilities of the GSARNN model and the other three models in three-dimensional space, revealing the advantages



and disadvantages of each model.

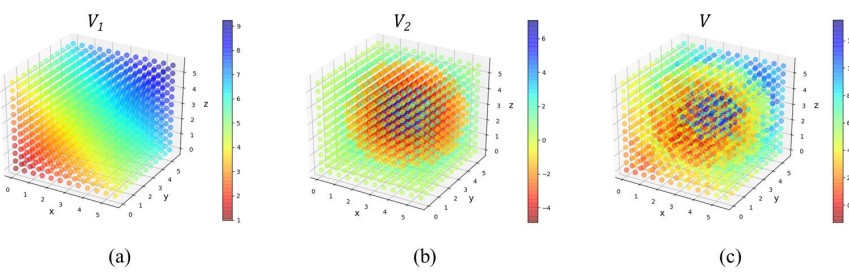

|  (a)  |  (b)  |  (c)  |

Figure 5. An example of a simulated dataset. (a) Component $V_1$ with a gradual change trend; (b) component $V_2$ with a mutation; (c) the simulated dataset combining $V_1$ and $V_2$.

### 3.1.2 Experiment implementation

According to the data division method, the simulated dataset is randomly divided into 10 equal parts for the cross-validation experiments. Each experiment has 1,555 data points in the training set and 173 data points in the validation set. The validation set interpolation results of each fold are merged to obtain the complete interpolated dataset.

Considering that the four-layered feedforward network is a simple but efficient network structure (Tamura and Tateishi, 1997), we design the GSARNN architecture with one input layer, two hidden layers, and one output layer. The number of neurons in the input layer and output layer is equal to the number of sample points in the training set. There is no standard method to determine the optimal number of neurons in two hidden layers. Instead, we determine the optimal number using a simple combination strategy proposed by Du et al. (2020). Table 1 lists the optimal network structure settings and hyperparameters of the GSARNN model in this case. The generalized spatial distance output by the GSDNN unit serves as the input for the GSARNN model, while the three-dimensional spatial Euclidean distance serves as the input for the three comparison methods.

Table 1. Network structure settings and hyperparameters of the GSARNN model in case one.

| GSDNN | Input | Hidden 1 | Hidden 2 | Output |
|---|---|---|---|---|
|  | 3 | 5 | 3 | 1 |
| GSARNN | Input | Hidden 1 | Hidden 2 | Output |
|  | 1555 | 512 | 512 | 1555 |
| Hyperparameters | $\alpha_{start}$ | $\alpha_{max}$ | Max Epoch | Batch Size | Dropout |
|  | 0.01 | 0.02 | 40000 | 64 | 0.75 |

### 3.1.3 Results

Under the same conditions, interpolation experiments are conducted on the three-dimensional simulated dataset 100 times using the IDW and OK methods and the SARNN and GSARNN models. The mean statistical indicator results of the cross-validation experiments are shown in Table 2. Compared with the traditional IDW and OK methods, the two neural network methods show significant improvements on all statistical indicators. The $R^2$ value of the SARNN model (0.8804) is 19.62% higher than that of the OK method (0.7360). After integrating the GSDNN unit, the $R^2$ increases by 5.95% to 0.9328 for the GSARNN, for an overall increase of 26.74%. In addition, the RMSE, MAE, and MAPE values of the OK method are 1.4466, 1.0040, and 92.69%, respectively, decreasing to 0.7298, 0.5280, and 40.99%, respectively, for the GSARNN. Among the four models, the GSARNN model has the best performance in all indicators.





Table 2. The mean statistical evaluation results of 100 experiments on the simulated dataset using the IDW, OK, SARNN, and GSARNN methods.

| Model | $R^2$ | RMSE | MAE | MAPE |
|--------|--------|--------|--------|--------|
| IDW | 0.7048 | 1.5295 | 1.0213 | 90.59% |
| OK | 0.7360 | 1.4466 | 1.0040 | 92.69% |
| SARNN | 0.8804 | 0.9736 | 0.6291 | 48.33% |
| GSARNN | 0.9328 | 0.7298 | 0.5280 | 40.99% |

Figure 6 shows the three-dimensional diagrams of the simulated dataset example in Fig. 5 and its corresponding cross-validation interpolation results generated by the four methods. Taking the simulated dataset as a reference, all four methods express the overall change trend, but the IDW and OK methods perform poorly in the mutation area, which presents as the weakening of the mutation trend and the existence of an obvious interpolation transition zone. The interpolation results of the SARNN and GSARNN models capture and display the mutation characteristics well, and the overall pattern is basically consistent with the simulated data.

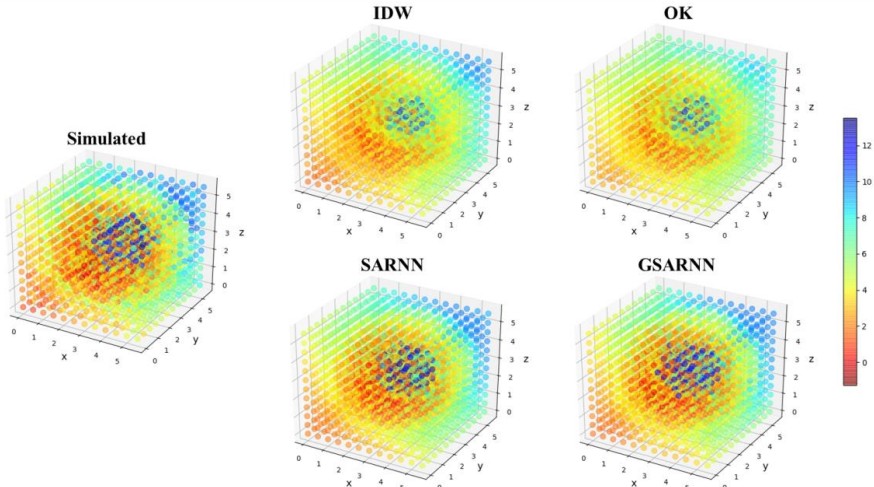

Figure 6. Three-dimensional diagrams of the simulated dataset example in Fig. 5 and its corresponding cross-validation interpolation results.

Figure 7 shows the detailed interpolation results of Fig. 6 in the form of section images, which are cut along the X–Y plane. In the IDW and OK method results, the low values of the mutation area are obviously overestimated, and the high values are underestimated. Moreover, under the influence of the central high value, unexpected imprints appear in the fourth and tenth layers, which reflects the limitations of traditional interpolation methods in handling mutation. The SARNN and GSARNN models largely restore the data characteristics of the simulated dataset, and the mutation is properly interpolated. Furthermore, the GSARNN model achieves more accurate results at the mutation interface.



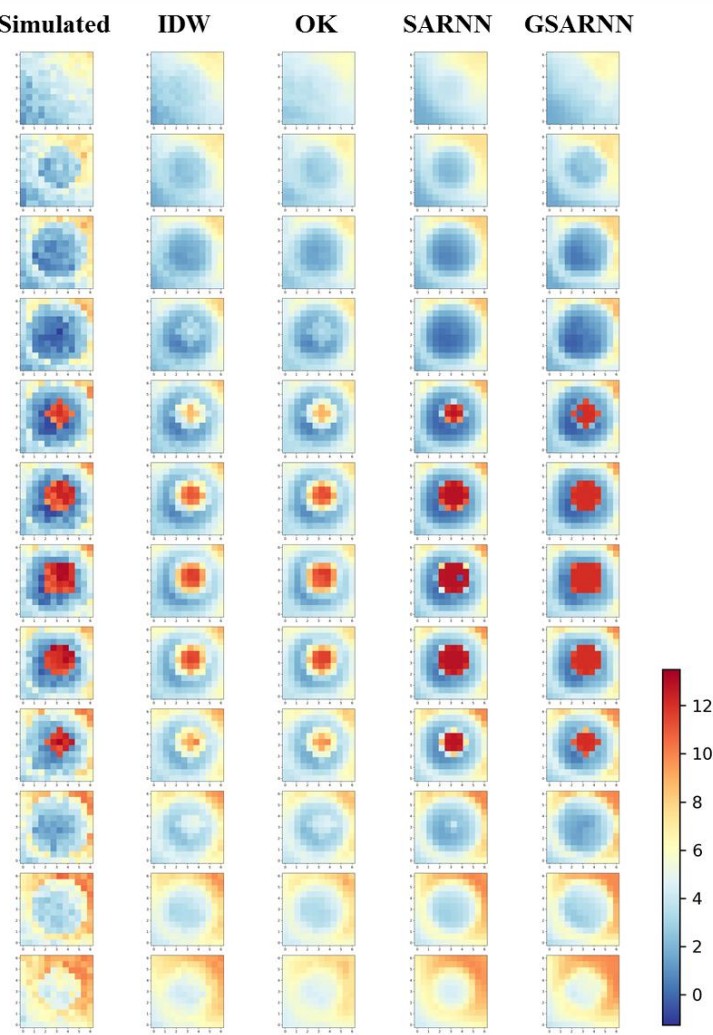

Figure 7. Section images cut along the X–Y plane of Fig. 6.

The real value and the cross-validation interpolation result values of the four models in Fig. 6 are drawn in line charts in Fig. 8. To evaluate the model performance in different value ranges, the line charts are drawn in ascending order of the real value, which is shown as a rising blue curve. The red line is connected by the model interpolation result points corresponding to the points in the simulated dataset, shown as a fluctuating broken line. In the median value area, the interpolation results of the four methods fluctuate relatively slightly near the real value. The IDW and OK method results show obvious low-value overestimation and high-value underestimation in a large range of low and high values, corresponding to both sides of the mutation interface. Limited by the interpolation mechanism and simplicity of traditional methods, it is difficult for them to interpolate elements containing mutation characteristics. However, the fluctuation amplitude and deviation degree of the OK method result are slightly smaller than those of the IDW method. The interpolation performances of the SARNN and GSARNN models in each value range are comparatively stable. Only a slight overestimation is observed in the low value area, but there are individual points with large deviations in the high value area. By contrast, the interpolating capacity of the GSARNN for mutant elements is significantly better than that of the SARNN.

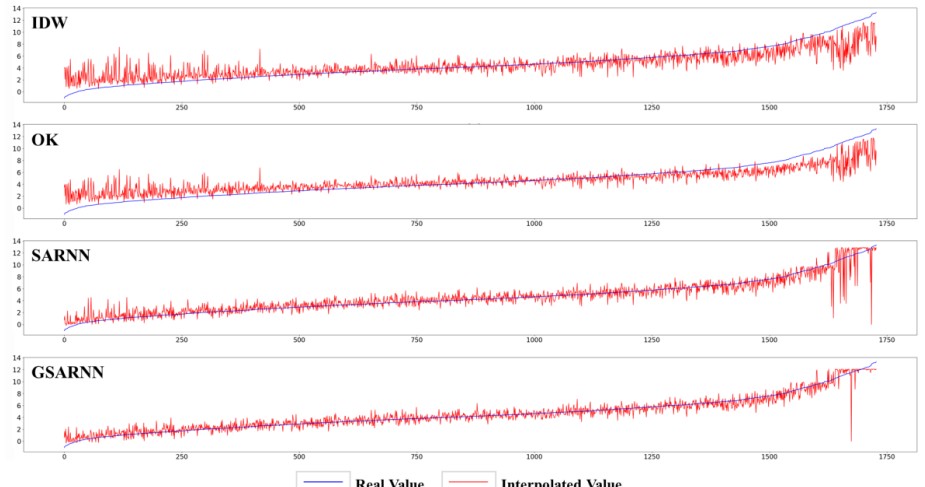

Figure 8. Line charts of real and interpolated values of the four models in Fig. 6.

**3.2 Case two: Argo dataset**

**3.2.1 Study area and dataset**

The second case uses the measured Argo ocean dataset. The study area is in the northern part of the western Pacific, which is located near the equator, and is one of the main sources of atmospheric water vapor. The sea–atmosphere interaction in this area is strong and exerts certain influences on natural phenomena such as El Niño (Jian and Jin, 2008); therefore, it is of practical significance to conduct research in this region. Water temperature is one of the most important oceanographic elements. Because the western Pacific is the divergent center of three major monsoon circulations, and multiple ocean currents

converge here, the seawater temperature in this area has a substantial impact on the natural environment. This case uses the sea temperature in the western Pacific as the interpolation object.

Three-dimensional temperature data were obtained from the Argo (Array for Real-time Geostrophic Oceanography) project, which was initiated to study global oceanic climate change. The Argo observation network has launched 3,000 profile buoys that measure the ocean temperature and salinity in the depth range of 2000 meters (Riser et al., 2016). Argo data have become

the main source of marine climate information and are widely used in marine and climate research (Liu et al., 2017). However, the Argo buoys are sparsely distributed, and the practical applications of the discrete data they collect are limited. Therefore, interpolating Argo data is necessary for generating a continuous data field and enhancing the practicability of the data products.

The data used in this case were obtained from China's Argo Real-time Data Center (http://www.argo.org.cn/). The data collection time is early August 2018, the space range is 0°–34°N, 115°–160°E, and the depth range is 0 to 1000 meters below

the sea surface. The data include measurements from 144 buoy stations and 1,944 monitoring items. The distribution of buoy stations on the sea surface is relatively uniform, as shown in Fig. 9.



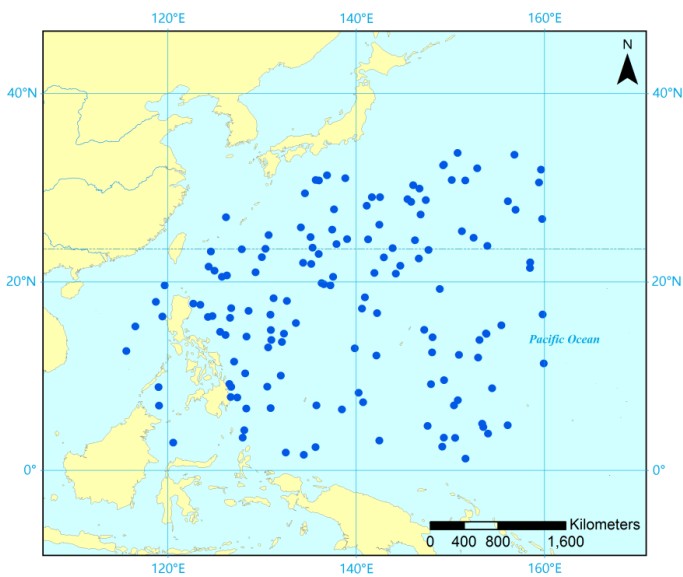

Figure 9. Distribution map of Argo buoy stations represented by blue points, 144 stations in total (the base map is from ESRI maps).

A three-dimensional visualization of the Argo dataset is shown in Fig. 10. The temperature field data in the western Pacific region are distributed regularly, with obvious and uniform variation trends and strong spatial correlation. Little temperature variation is observed in the longitudinal direction. In the latitudinal direction, the boundary between the low temperature region and the high temperature region sinks obviously, and the overall temperature increases with increasing latitude from 0° to 35°N. In the vertical direction, the temperature decreases gradually with increasing water depth. The mean, minimum, maximum, and standard deviation of the Argo temperature dataset are shown in Table 3.

**Argo**

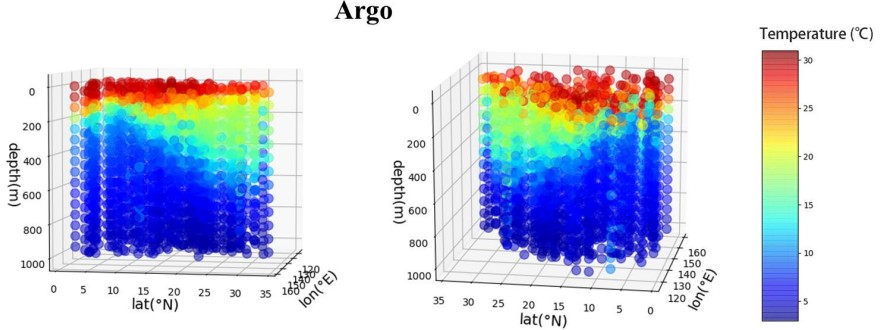

Figure 10. Three-dimensional visualization of the Argo temperature dataset.

Table 3. Basic statistics of the Argo temperature dataset with 1944 monitoring points in total.

| Statistics | Mean | Minimum | Maximum | Standard Deviation |
|---|---|---|---|---|
| $V$ (℃) | 12.1958 | 3.6230 | 29.9890 | 7.6791 |

This case compares the interpolation abilities of the GSARNN model and the other three models in three-dimensional space using real temperature data collected by Argo buoys.

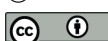

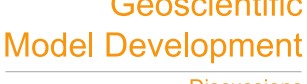

### 3.2.2 Experiment implementation

The model details are determined in a similar way to case one. The optimum network structure settings and hyperparameters of the GSARNN model for case two are listed in Table 4.

Table 4. Network structure settings and hyperparameters of the GSARNN model in case two.

| GSDNN | Input | Hidden 1 | Hidden 2 | Output |
|---|---|---|---|---|
| | 3 | 5 | 3 | 1 |
| GSARNN | Input | Hidden 1 | Hidden 2 | Output |
| | 1749 | 512 | 256 | 1749 |
| Hyperparameters | $\alpha_{start}$ | $\alpha_{max}$ | Max Epoch | Batch Size | Dropout |
| | 0.005 | 0.01 | 30000 | 64 | 0.75 |

### 3.2.3 Results

Under the same conditions, interpolation experiments are conducted on the three-dimensional measured Argo dataset using the IDW, OK, SARNN model, and GSARNN model methods. The statistical indicators for the cross-validation experiments are shown in Table 5. In contrast to the simulated dataset of case one, the values of the Argo dataset mainly change in a gradual manner, which is relatively simple. Therefore, all four methods achieve satisfying interpolation experimental results on the

whole. However, we notice that certain differences of interpolation accuracy exist in local high value region ($V \geq 20°C$). From OK to SARNN to GSARNN, local $R^2$ increases from 0.7928 to 0.8784 to 0.8949, representing increases of 10.80% and 12.88%, respectively. The RMSE, MAE, and MAPE values of the SARNN are slightly lower than those of the traditional methods. After integrating the GSDNN unit, the three indicators decrease significantly from 0.9282, 0.6056, and 5.80% for the SARNN to 0.7169, 0.4653, and 4.62% using the GSARNN. Among the four methods, the GSARNN model has the best

performance in all indicators.

Table 5. The statistical evaluation results of the Argo temperature dataset experiments using the IDW, OK, SARNN, and GSARNN methods.

| Model | Local $R^2$ (region where $V \geq 20°C$) | Global $R^2$ | RMSE | MAE | MAPE |
|---|---|---|---|---|---|
| IDW | 0.5986 | 0.9743 | 1.2316 | 0.7725 | 6.70% |
| OK | 0.7928 | 0.9815 | 1.0442 | 0.6496 | 5.55% |
| SARNN | 0.8784 | 0.9854 | 0.9282 | 0.6056 | 5.80% |
| GSARNN | 0.8949 | 0.9913 | 0.7169 | 0.4653 | 4.62% |

Figure 11 shows three-dimensional diagrams of the measured Argo temperature dataset and the cross-validation interpolation results generated by the four methods. Taking the real dataset as a reference, the four models restore the data features in most

areas, which is consistent with the statistical indicator results, with small differences in some details. The IDW method evidently underestimates the temperature in low depth areas, which may be because its interpolation mechanism can produce large errors at the edge points of a given space. In the OK results, the coexistence of underestimation and overestimation around the sea surface is observed, indicating that the OK method also has some limitations in edge-area interpolation. The SARNN and GSARNN models slightly overestimate the bottom area. Further quantitative analysis is needed to elucidate more

details of the interpolation experiment results.

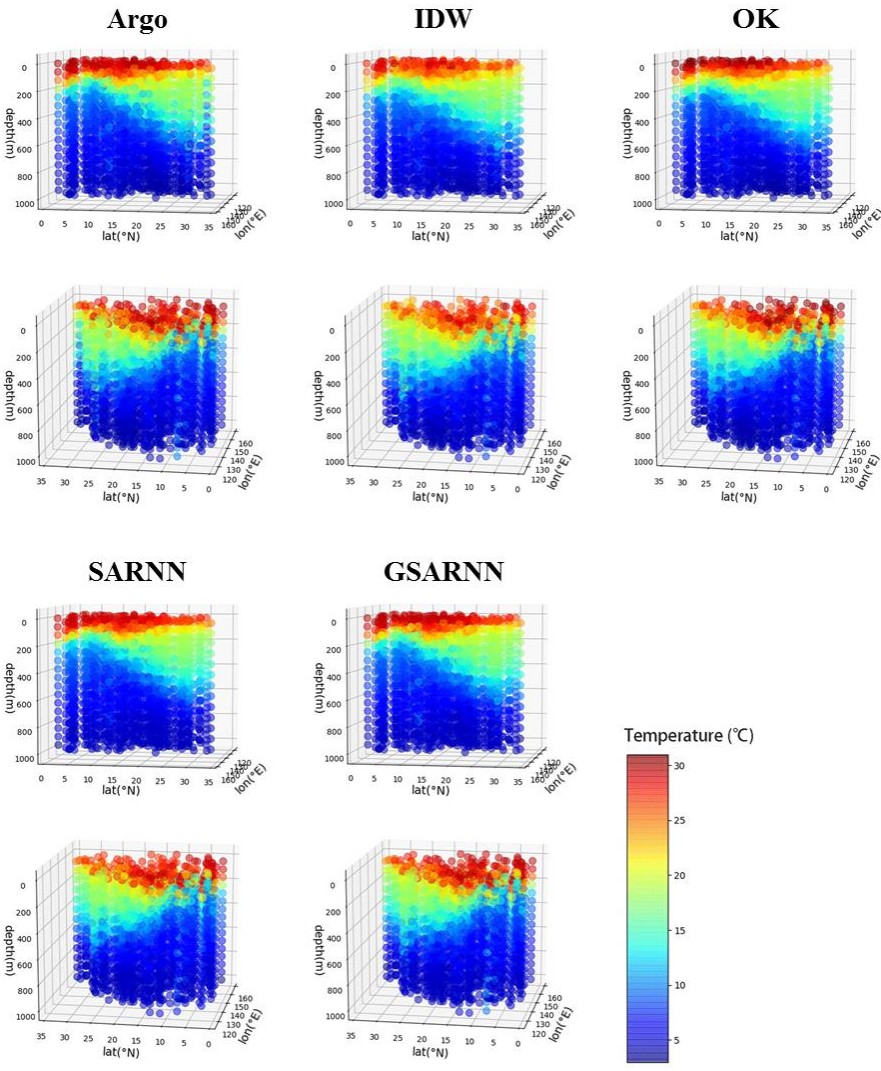

Figure 11. Three-dimensional diagrams of the real dataset and the cross-validation interpolation results.

The cross-validation interpolation result values of the four models and the real values are respectively drawn as line charts and scatter diagrams, as shown in Fig. 12(a) and Fig. 12(b). In the low value area, the fluctuations of the four models are generally small. Several points with large errors are in similar positions for all models, indicating that there may be individual outliers in the dataset; however, the GSARNN model has the strongest ability to minimize these errors. In addition, the IDW, SARNN, and GSARNN methods marginally overestimate the lowest value. Entering the median area, the fluctuation of the four models begins to increase gradually; IDW produces the highest amplitude, followed in descending order by the OK, SARNN, and GSARNN methods. The GSARNN method avoids potential large errors in several positions to the greatest extent. In the high value area, the performances of the four models are more distinct. The IDW method underestimates the high values across a large range, the SARNN model slightly underestimates them, the OK method fluctuates around the real values, and the GSARNN model hovers within a narrow range. The information conveyed by the scatter diagrams is consistent with the line charts. The scatter diagrams show that the scatter points of all four methods are concentrated around the diagonal, and the



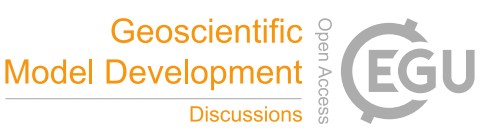

trend lines almost coincide with the standard trend line. Among them, the performance of the GSARNN is quantitatively best.

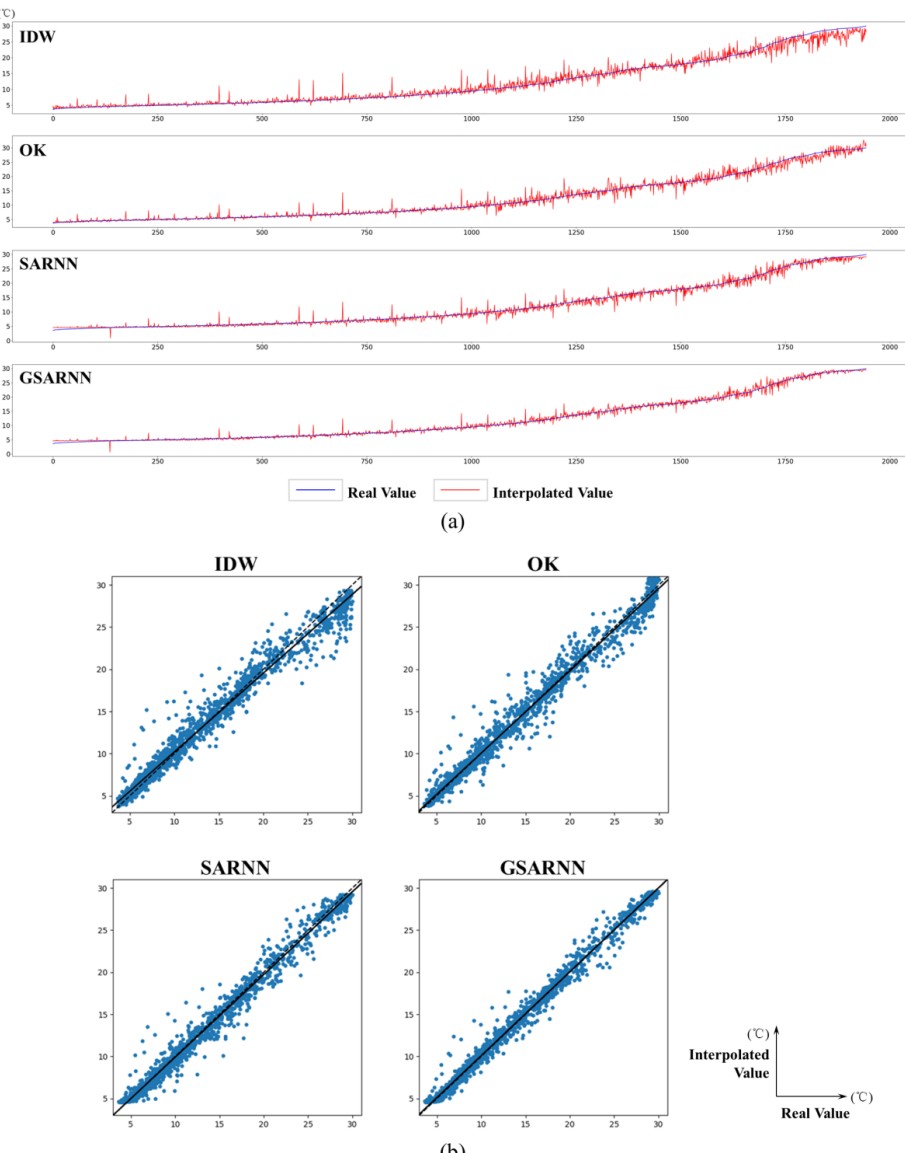

Figure 12. The quantitative analysis results of the four models. (a) line charts in ascending order of real value; (b) scatter diagrams with trend lines.

5     To compare the visual performance and effects of the four methods for practical interpolation applications, we interpolate and render horizontal sections at 100 m depth intervals in this area. Each method generates nine sections of 0–800 m depth, as shown in Fig. 13. The four methods produce similar interpolation results on the overall pattern, but there are great differences in detail. Due to the sparsity of the sampled points, the points closer to the section have a more prominent impact than the distant points in the interpolation results of the IDW method, producing many noticeable speckles on the interpolation surface.

10    The OK method uses the statistical calculation process to fit the spatial features to a certain extent, alleviating the speckle problem; however, uneven color bands with abrupt color changes can still be observed. The SARNN and GSARNN models



fit the continuous temperature field characteristics using the same set of sparse Argo temperature data. The overall change trend of the interpolated sections is consistent with the traditional methods but is significantly smoother and more uniform, reflecting the actual temperature field characteristics. Compared with the SARNN, the GSARNN presents richer details on the basis of smoothness, more exhaustively describing the ocean temperature field characteristics, showing the qualitatively best

5   performance.

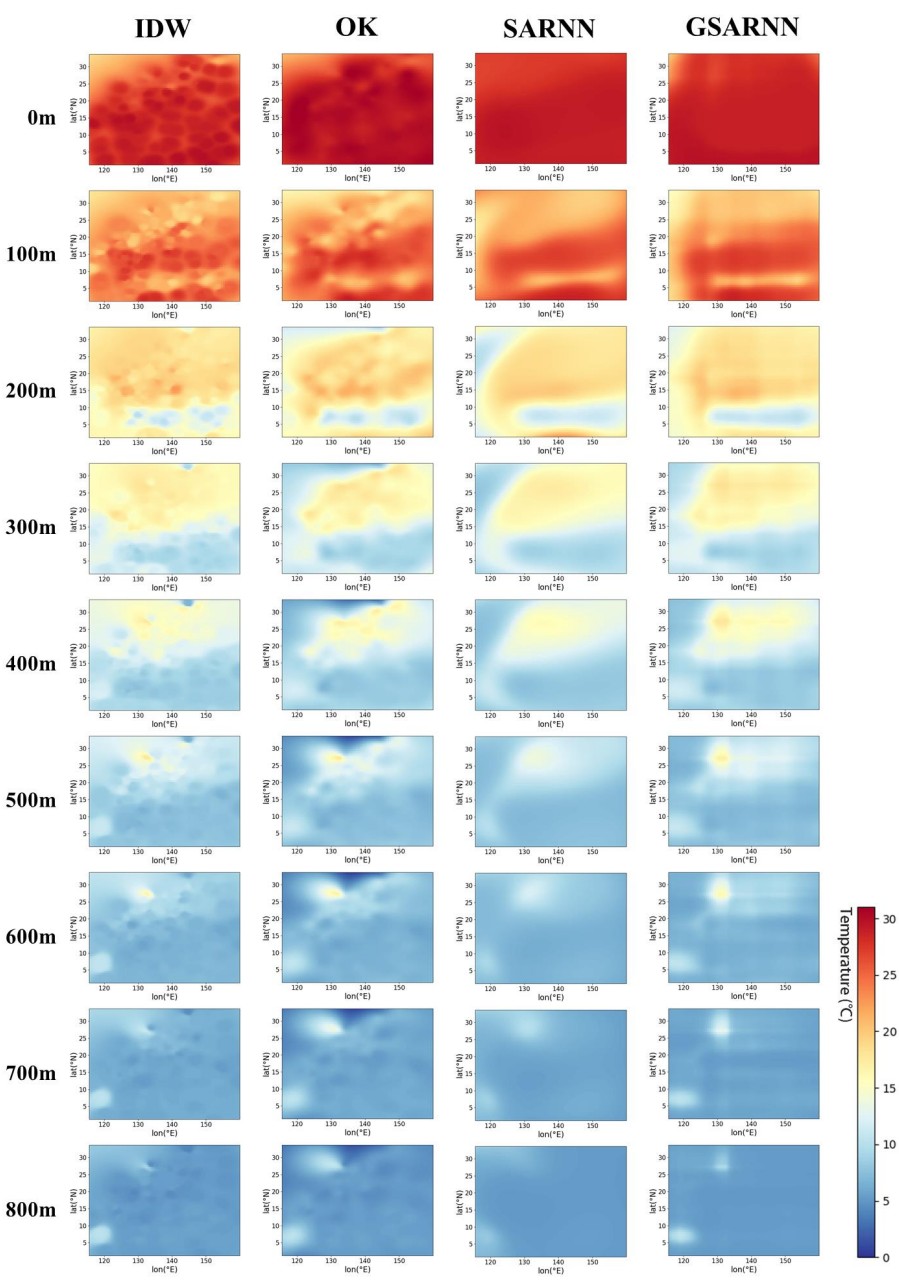

Figure 13. Comparisons of interpolated horizontal sections at 100 m depth intervals generated by the four methods.





## 4. Discussion

In case one, the comparison between two traditional interpolation methods and two neural network-based methods demonstrates that introducing neural networks for powerful nonlinear fitting improves interpolation performance, enabling the adequate extraction and construction of complex change characteristics of spatial elements such as mutation. The comparison

of the SARNN and GSARNN models shows that deconstructing and remodeling the expression and solution of spatial distances, and subsequently applying the generalized expression in interpolation calculations, enables the model to mine and restore the characteristics of the original data to the greatest extent, effectively improving the interpolation accuracy and optimizing the interpolation result.

In case two, the section interpolation prediction performance of the four methods varies considerably. The spatial distribution

of the Argo buoys is sparse, uneven, and irregular, which is common in most practical interpolation scenarios. When interpolating such datasets, the traditional methods tend to produce dominant weights on the points adjacent to the point to be interpolated, which may lead to disproportionate regional impacts of specific sample points around them, resulting in uneven speckles and bands. Traditional methods lack the global consideration of the comprehensive effect of all sample points on the interpolation area. In contrast, the GSARNN model incorporates the raw coordinate vectors as the network input and fits the

generalized spatial distances in the three-dimensional spatial element field, extracting more global and detailed data features, generating interpolation results that are more consistent with reality.

In summary, in case one, we test the quantitative interpolation performance of the four methods on a dataset with complex characteristics; in case two, we examine the qualitative performance of the four methods in a practical interpolation application. The experiment results indicate that traditional methods are sensitive and dependent on the spatial distribution and data

characteristics of the sampled points. By applying the concepts of neural networks, spatial autoregression, and generalized spatial distances to three-dimensional spatial interpolations, the GSARNN model can effectively optimize the interpolation result and improve the adaptability of interpolation methods in various scenarios.

## 5. Conclusion

In this study, we focus on the integration of interpolation and neural network model in three-dimensional space, in which the

spatial elements possess complex characteristics. To improve the interpolation effect, we remodel the expression and solution of spatial distances and spatial weights—two critical elements in spatial interpolation. For spatial distance, we employ the generalized spatial distance expression and propose a GSDNN unit to adaptively generate the generalized spatial distance, replacing the conventional Euclidean spatial distance as the interpolation network input. For spatial weight, we construct the GSARNN model by integrating the GSDNN unit into the SARNN model. Exploiting the powerful feature extraction and

nonlinear fitting abilities of neural networks, we can realize accurate spatial weight calculations.

Experiments are conducted on two three-dimensional cases: a simulated case and a real Argo temperature case. The GSARNN model is compared with the traditional IDW and OK methods and the advanced SARNN model. The experiment results indicate that the GSARNN model achieves the best interpolation performance among the four methods, especially on the complex three-dimensional spatial dataset with discontinuous features and sparse and irregular distribution. The GSARNN

model can effectively extract subtle spatial correlation characteristics and accurately fit the spatial weights, adapting well in three-dimensional space.

The GSARNN can perform spatial interpolation with high accuracy at the cost of longer model training and calculation time. Therefore, the GSARNN is more suitable for interpolation scenarios with complex characteristics and strict demands on the



result quality. For interpolation tasks with relatively simple spatial characteristics and specific requirements for efficiency, traditional methods may be a better choice.

In the future, we plan to consider the time dimension in addition to the spatial dimension to develop an accurate spatiotemporal data interpolation model. Meanwhile, based on the interpolation dependent variable, the relevant regression variable factors

can be introduced for further interpolation statistical analyses.

## Code and data availability

Simulated data, Argo temperature data and codes used in the study are available at https://figshare.com/s/8d5e4b6e5b74cc1e 0bc1.

## Author contributions

JZhan, SW, JZeng, ZD developed the model. JZhan, SW, JQ implemented the model and conducted the experiments. JZhan,

SW, MQ, YW contributed to the planning and discussions and to the writing of the article.

## Competing interests

The authors declare that they have no conflict of interest.

## Acknowledgments

## Financial support

This work was supported by the National Natural Science Foundation of China [No. 41922043, No. 41871287, No. 42001323]; the National Key Research and Development Program of China [No. 2021YFB3900902, No. 2018YFB0505000]; and the Provincial Key R&D Program of Zhejiang [No.2021C01031].

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
