# Peer review of "A generalized spatial autoregressive neural network (GSARNN) method for three-dimensional spatial interpolation"

_Geoscientific Model Development, 2022_

## Author Response (AR1)

**Chief Editor**

Dear Editor:

We would like to express our great gratitude for you to give us the opportunity to revise our manuscript "A generalized spatial autoregressive neural network (GSARNN) method for three-dimensional spatial interpolation". We have made a revision of the manuscript according to the comments and suggestions from you. These comments are all valuable and very helpful for improving our paper. We sincerely hope that our revisions would meet your requirements. Please don't hesitate to contact us if you have any problems about the response.

**Comment 1:** First, your code in FigShare does not include a license. If you do not include a license, despite what you state in the README file, the code is not "open-source"; it continues to be your property. Therefore, you should add a free software/open-source (FLOSS) license to the repository and mention it in the code. We recommend the GPLv3. You only need to include the file 'https://www.gnu.org/licenses/gpl-3.0.txt' as LICENSE.txt with your code. Also, you can choose other options, for example, the GPLv2, Apache License, MIT License, etc.
**[Response]**: Thanks for your very helpful comment. According to your suggestion, we have adopted the GPLv3 license to our code and data in FigShare. A text file named "LICENSE.txt" obtained from 'https://www.gnu.org/licenses/gpl-3.0.txt' has been upload to the original FigShare link in Section code and data availability.

**Comment 2:** Also, for your work, you use TensorFlow. This is not clear in the manuscript, and it is an important detail. Therefore, please, add more detail in the Methods section of the text about the specific software used and their versions.
**[Response]**: Thanks for your very helpful comment. We are sorry for neglecting this important detail. The GSARNN and SARNN models are implemented using TensorFlow-GPU 1.13.0 and Python 3.5.4. This information has been added in the section of experiment implementation (Section 3.1.2, Paragraph 2), which we

consider it may be more suitable than adding it in Methods section.

*"The GSARNN and SARNN models are implemented using TensorFlow-GPU 1.13.0 and Python 3.5.4."*

**Reviewer 1**

Dear Referee:

Thanks very much for your great support and constructive suggestions with regard to our manuscript. These comments are all valuable and very helpful for revising and improving our paper, as well as the important guiding significance to our researches. We have made our best efforts to improve our paper very carefully following your comments and suggestions. Our point by point response to the comments are given below. We hope the revised manuscript will be acceptable to your requirements. If still there are concerns, we will be happy to take care once we hear from you.

**Major improvements:**

**Comment 1:** GSDNN is critical in enhancing the GSARNN model's fitting accuracy. This highlights the significance of obtaining the correct spatial correlation between sample points by taking into account both the dataset's local and global qualities. I'm simply wondering if using GSDNN with traditional approaches will result in better interpolation results.

**[Response]**: Thanks for your comment. The GSDNN unit plays an important role in GSARNN model, taking variation characteristics of geographical elements in different directions into account. However, since there is no recognized true value of generalized spatial distance for training process, applying GSDNN unit to traditional methods will make the calculation process impossible to carry on. Therefore, the GSDNN unit can only be embedded in the NN method and participates in its overall training and calculation process. In other words, the generalized spatial distance is determined by the spatial characteristics of the elements to be interpolated, owning a

specific connotation based on specific context of spatial elements. Some additional explanations have been added in the last paragraph of Section 2.3.1.

*"Note that since there is no recognized true value of generalized spatial distance for training process, the GSDNN unit can only be embedded in the neural network-based method and participates in its overall training and calculation process. In other words, the generalized spatial distance is determined by the spatial characteristics of the elements to be interpolated, owning a specific connotation based on specific context of spatial elements."*

**Comment 2:** Section 2.1.2 should go over the kriging approach in greater detail. Please explain how the weight coefficient $\lambda_i$ is calculated.

**[Response]**: Thanks for your helpful advice. We have properly supplemented the interpolation calculation process of the Kriging method, including how the weight coefficient $\lambda_i$ is calculated. However, since the focus of this paper is on the GSARNN model, the derivation processes of some formulas in Kriging method have been omitted to make the length of Section 2.1.2 not too long.

*Kriging can be expressed as:*

$$z^*(x_0) = \sum_{i=1}^{n} \lambda_i z(x_i) \qquad (2)$$

*where $z^*(x_0)$ is the predicted value, and $\lambda_i$ and $z(x_i)$ are, respectively, the weight coefficient and observed value of point $i$.*

*Kriging methods involve the calculation of the weight coefficient $\lambda_i$, for which the key is to satisfy the unbiasedness and optimality. Unbiasedness means that $z^*(x_0)$ is the unbiased estimate of $z(x_i)$, that is:*

$$E[z^*(x_0) - z(x)] = 0 \qquad (3)$$

*which can derive the following constraints on $\lambda_i$:*

$$\sum_{i=1}^{n} \lambda_i = 1 \qquad (4)$$

*Optimality means that $z^*(x_0)$ is the optimal estimate of $z(x_i)$, and the variance between the predicted value of the unsampled points and the estimated value of the*

*observed points is the smallest, that is:*

$$\min_{\lambda_i} Var(z^*(x_0) - z(x))$$ (5)

*Define the cost function and try to figure out a set of weights $\lambda_i$ that satisfy unbiasedness and minimize the cost function. Finally, the following equation set can be derived:*

$$\begin{cases} \sum_{i=1}^{n} r_{ij}\lambda_i = r_{j0}, j = 1,2,...,n \\ \sum_{i=1}^{n} \lambda_i = 1 \end{cases}$$ (6)

*where $r_{ij}$ is the semi-variogram between point $i$ and point $j$, which can be expressed as:*

$$r_{ij} = \sigma^2 - Cov(z_i, z_j) = \frac{1}{2}E[(z_i - z_j)^2]$$ (7)

*where $\sigma^2$ is the variance of $z(x)$, which is a constant in OK. $r_{ij}$ can be simply determined by $z_i$ and $z_j$. Kriging assumes that there is a functional relationship between $r_{ij}$ and $d_{ij}$ (the distance between point $i$ and point $j$) and $\frac{n(n-1)}{2}$ $(r, d)$ pairs can be generated by taking any two sampled points from the dataset. We can use linear, power, gaussian, spherical or exponential model to fit the relationship between $r_{ij}$ and $d_{ij}$. Using the fitted function, we can calculate $r_{j0}$ through $d_{j0}$. Thereby, the optimal weight set $\lambda_i$ in Formula 6 can be solved.*

**Comment 3:** The phrase "weight matrix" appears for the first time on Page 5, Line 6. Please provide some context.

**[Response]:** Thanks for your instructive advice. The logic of space weight description in the original manuscript is not coherent enough, which will make some readers confused. The overall presentation of this part (Section 2.2) has been properly reorganized to make it easier to understand.

*It should be noted that there is a weight $w_{ii}$ in the vector $\boldsymbol{w}_i$ which represents the spatial weight of point $i$ to itself. To avoid overfitting, this weight should be set to 0:*

$$w_{ij} = \begin{cases} f(d_{i1}^s, d_{i2}^s, \cdots, d_{in}^s)_j, & i \neq j \\ 0, & i = j \end{cases} \tag{9}$$

*The spatial weights of all sampled point pairs can be expressed by an $n * n$ weight matrix $\boldsymbol{W}$. According to Formula 9, the weights on the diagonal of $\boldsymbol{W}$ should be reset to 0. Therefore, $\boldsymbol{W}$ can be defined as:*

$$\boldsymbol{W} = \boldsymbol{\rho} * \boldsymbol{K} \tag{10}$$

*where $\boldsymbol{\rho}$ is the spatial weight component, and $\boldsymbol{K}$ is the standard weight component, which ensures that the neural network weight is independent of the point itself in the training process. $k_{ij}$ in $\boldsymbol{K}$ can be expressed as:*

$$k_{ij} = \begin{cases} 1, & i \neq j \\ 0, & i = j \end{cases} \tag{11}$$

**Comment 4:** You say in section 2.3.3 that you utilize variable learning rate for network training and explain how it changes during the training process. You should also consider the benefits of this customized learning rate.

**[Response]:** Thanks for your very helpful advice. The learning rate starts from $\alpha_{start}$ and increases gradually at the rate of $k_1$ until $\alpha_{max}$. A relatively small initial learning rate can prevent excessive fluctuation and convergence obstacle and the following increment of leaning rate can avoid the convergence rate at the early stage of the training process being too low. The maximum learning rate is maintained for $n$ epochs. At this stage, the model can stably learn the spatial characteristics of the elements. The learning rate then gradually decreases exponentially at the rate of $k_2$, ensuring that the model can sufficiently converge near the optimal position. The description of the customized learning rate benefits has been added in Section 2.3.3, Paragraph 3, which makes this strategy more reasonable.

*"where $\alpha_{start}$ is the initial learning rate, which increases gradually at the rate of $k_1$ until $\alpha_{max}$. A relatively small initial learning rate can prevent excessive fluctuation and convergence obstacle and the following increment of leaning rate can avoid the convergence rate at the early stage of the training process being too low. The maximum learning rate is maintained for $n$ epochs. At this stage, the model can stably learn the spatial characteristics of the elements. The learning rate then*

*gradually decreases exponentially at the rate of $k_2$, ensuring that the model can sufficiently converge near the optimal position. The change of the learning rate throughout the training process is shown in Fig. 3.”*

**Comment 5:** The difference between all of these interpolation solutions is difficult to notice in Figure 11. I recommend graphing the difference between the interpolated and real values (interpolation error) and modifying the color scheme accordingly.

**[Response]**: Thanks for your very instructive advice. According to your recommendation, we graph the cross-validation interpolation results as well as the interpolation errors of the four methods as Figure 10 (Figure 11 in the original manuscript). The difference of the interpolation performances between the four methods can be more easily distinguished now. The caption and paragraph corresponding to this figure have also been appropriately revised.

*“Figure 10 shows three-dimensional diagrams of the cross-validation interpolation results generated by the four methods and their corresponding interpolation errors. In interpolation error diagrams, red represents overestimation and blue represents underestimation. Taking the real dataset in Figure 9 as a reference, the four models restore the data features in most areas, which is consistent with the statistical indicator results, with small differences in some details. The IDW method evidently underestimates the temperature at shallow depths, which may be because its interpolation mechanism can produce large errors at the edge points of a given space. In the OK results, the coexistence of underestimation and overestimation around the sea surface is observed, indicating that the OK method also has some limitations in edge-area interpolation. The SARNN and GSARNN models slightly overestimate the temperature of bottom area. The error of GSARNN is generally smaller than that of SARNN. Further quantitative analysis is needed to elucidate more details of the interpolation experiment results.”*

[Figure]

*Figure 10. Three-dimensional diagrams of the cross-validation interpolation results and interpolation errors.*

**Comment 6:** It is unclear what the x-axis signifies in Figures 8 and 12. Please include some text and figure descriptions.

**[Response]:** Thanks for your helpful advice. The x-axis in Figures 8 and Figure 12 represents the identifier number of each sampled point after they are sorted in ascending order of real value. Without an explanation, readers may feel confused. Some descriptions have been added in Figure 8 and Figure 12.

[Figure]

*Figure 7. Line charts of real and interpolated values of the four models in Fig. 5.*

[Figure]

*Figure 11. The quantitative analysis results of the four models. (a) line charts in ascending order of real value; (b) scatter diagrams with trend lines.*

**Minor suggestions:**

**Comment 1:** In section 3.1.1, from Line 19 to Line 22, I suggest not overly emphasizing the benefits of this experiment using the simulated data. This point has

been mentioned in earlier paragraphs.

[Response]: Thanks for your helpful advice. Redundant statements have been removed. In order to keep Section 3.1.1 consistent with Section 3.2.1, a sentence is added at the end of Section 3.1.1.

*"This case compares the interpolation abilities of the GSARNN model and the other three models in three-dimensional space using the simulated dataset above."*

**Comment 2:** In Formula 10, since $k_{ij}$ is 1 for the situation $i \neq j$, then should the off-diagonal elements simply be written as $\rho_{ij}$?

[Response]: Thanks for your helpful advice. The issue has been corrected.

$$W = \rho * K = \begin{bmatrix} \rho_{11} & \rho_{12} & \cdots & \rho_{1n} \\ \rho_{21} & \rho_{22} & \cdots & \rho_{2n} \\ \vdots & \vdots & \ddots & \vdots \\ \rho_{n1} & \rho_{n2} & \cdots & \rho_{nn} \end{bmatrix} * \begin{bmatrix} 0 & 1 & \cdots & 1 \\ 1 & 0 & \cdots & 1 \\ \vdots & \vdots & \ddots & \vdots \\ 1 & 1 & \cdots & 0 \end{bmatrix} = \begin{bmatrix} 0 & \rho_{12} & \cdots & \rho_{1n} \\ \rho_{21} & 0 & \cdots & \rho_{2n} \\ \vdots & \vdots & \ddots & \vdots \\ \rho_{n1} & \rho_{n2} & \cdots & 0 \end{bmatrix} \tag{13}$$

**Comment 3:** Multiline formulas, such as Formula 6, Formula 19, Formula 30, should be left aligned. The "int" in Formula 26 and Formula 27 seems to be redundant.

[Response]: Thanks for your helpful advice. Formula 6, Formula 19, Formula 30 have been corrected to be left aligned. The "int" in Formula 26 and Formula 27 is to take the integer part of the operation result, which is the coordinate of the point in the cube.

$$w_{ij} = \begin{cases} f(d_{i1}^s, d_{i2}^s, \cdots, d_{in}^s)_j, & i \neq j \\ 0, & i = j \end{cases} \tag{9}$$

$$\alpha = \begin{cases} \alpha_{start} + k_1 epoch, & epoch < epoch_{up} \\ \alpha_{max}, & epoch \in [epoch_{up}, epoch_{down}] \\ k_2^{epoch} \alpha_{max}, & epoch > epoch_{down} \end{cases} \tag{22}$$

$$V_2 = \begin{cases} 1 - \dfrac{1}{36}(9-(3-x_i)^2)(9-(3-y_i)^2)(9-(3-z_i)^2), & dist[(x_i, y_i, z_i), (3,3,3)] \in [1.5,3] \\ 1 + \dfrac{1}{36}(9-(3-x_i)^2)(9-(3-y_i)^2)(9-(3-z_i)^2), & dist[(x_i, y_i, z_i), (3,3,3)] \notin [1.5,3] \end{cases} \tag{33}$$

**Comment 4:** Page 2, Abstract, Line 3: "which is one of the most important" to "a fundamental".

[Response]: Thanks. The issue has been corrected.

**Comment 5:** Page 2, Abstract, Line 10: "compared" to "compared with traditional methods".

[Response]: Thanks. The issue has been corrected.

**Comment 6:** Page 2, Line 17: "continuous data" to "continuous field".

[Response]: Thanks. The issue has been corrected.

**Comment 7:** Page 2, Line 32: "They" to "These methods".

[Response]: Thanks. The issue has been corrected.

**Comment 8:** Page 3, Line 7: "Zeng et al." to "In particular, Zeng et al.".

[Response]: Thanks. The issue has been corrected.

**Comment 9:** Page 3, Line 23: "by combining the GSDNN unit with the SARNN to integrate generalized distances into the spatial interpolation method, we developed" to "by combining the GSDNN unit with the SARNN, we integrated generalized distances into the spatial interpolation method and developed".

[Response]: Thanks. The issue has been corrected.

**Comment 10:** Page 4, Line 14: "deposit reserves" to "mineral deposit predication".

[Response]: Thanks. The issue has been corrected.

**Comment 11:** Page 6, Line 6: "speed" to "rate".

[Response]: Thanks. The issue has been corrected.

**Comment 12:** Page 9, Line 15: "the feature extraction and fitting ability of the GSARNN model are fully and persuasively tested" to "we can fully test the feature extraction and fitting ability of the GSARNN model".

[Response]: Thanks. The issue has been corrected.

**Comment 13:** Page 9, Line 17: "the most authentic" to "the authentic".

**[Response]**: Thanks. The issue has been corrected.

**Comment 14:** Page 10, Line 4: "sudden change" to "sudden variation".

**[Response]**: Thanks. The issue has been corrected.

**Comment 15:** Page 10, Line 17: "adds" to "imposes".

**[Response]**: Thanks. The issue has been corrected.

**Comment 16:** Page 10, Line 18: "ε" to "The term ε".

**[Response]**: Thanks. The issue has been corrected.

**Comment 17:** Page 11, Line 25: "has" to "achieves".

**[Response]**: Thanks. The issue has been corrected. The similar issue in Section 3.2.3 has also been corrected.

**Comment 18:** Page 16, Line 21: "in low depth area" to "at shallow depths".

**[Response]**: Thanks. The issue has been corrected.

**Comment 19:** Page 17, Line 5: "indicating that there may be individual outliers" to "indicating the presence of potential outliers".

**[Response]**: Thanks. The issue has been corrected.

**Reviewer 2**

Dear Referee:

Thanks very much for your great support and constructive suggestions with regard to our manuscript. These comments are all valuable and very helpful for revising and improving our paper, as well as the important guiding significance to our researches. We have made our best efforts to improve our paper very carefully following your

comments and suggestions. Our point by point response to the comments are given below. We hope the revised manuscript will be acceptable to your requirements. If still there are concerns, we will be happy to take care once we hear from you.

**Comment 1:** The authors only list the parameters of GSARNN model in detail in the manuscript. The configurations of traditional methods, such as the p value of IDW and the variation function adopted in Kriging, should also be mentioned in the comparison experiments.

**[Response]**: Thanks for your very helpful advice. The power parameter of IDW method is 4 in two cases. In Kriging method, we adopt the gaussian model to fit the functional relationship between the semi-variogram and the spatial distance, which turns out to be the optimal variation function model among linear, gaussian, spherical and exponential models. Some explanations have been added in Section 3.1.2, Paragraph 2.

*"Besides, the power parameter of IDW method is 4, and in Kriging method, we adopt the gaussian model to fit the functional relationship between the semi-variogram and the spatial distance, which turns out to be the optimal variation function model among linear, gaussian, spherical and exponential models."*

**Comment 2:** The results of case 2 turn out that the neural network-based models generate smoother spatial patterns than traditional methods. I wonder if that is worth discussing.

**[Response]**: Thanks for your very helpful advice. As you mentioned, the interpolation results of neural network-based models exhibit smoother spatial patterns with less noise than those of traditional methods. This indicates that neural network-based models can greatly reduce the influence of local extreme points on the points to be interpolated and acquire quite reasonable distributions of the geospatial elements through the non-linear fitting ability of neural networks. Some discussion has been added in Section 4 Discussion, Paragraph 2.

*"In contrast, neural network-based models generate smoother interpolation surface*

*than traditional methods. This indicates that neural network-based models can greatly reduce the influence of local extreme points on points to be interpolated and acquire quite reasonable spatial patterns of geospatial elements exploiting the non-linear fitting ability of neural networks."*

**Comment 3:** I think the point of how long it takes to run the model deserves more discussion in the manuscript. The authors briefly mention this as a limitation in the conclusion section, but some basic statistics on how long it takes would be a helpful addition.

**[Response]:** Thanks for your very insightful advice. As you mentioned, the model complexity of GSARNN is considerably higher than traditional methods. Nonetheless, compared with multifarious models in the fields of neural networks and deep learning, the structure of GSARNN with a few hidden layers is relatively lightweight, so its training and calculation efficiency can be quite high. The GSARNN model usually converges to the optimal state within 15-20 minutes in our cases since it can take advantage of mighty parallel computing capabilities of GPU units and distributed computing structures to accelerate the training process. Although the efficiency of Kriging method is better than GSARNN model, under the same condition, it still takes about 10 minutes to fit the functional relationship between the semi-variogram and the distance using "pykrige". However, as the number of sampled points increases, the number of input neurons and output neurons of the GSARNN will also increase, resulting in the expansion of network parameters and the extension of training time inevitably. How to maintain a stable and acceptable training time given different sample data volumes is an important problem to be tackled in further researches.

Some discussions have been added in the end of Section 3.1.3 (an additional paragraph) and Section 5 Conclusion, Paragraph 4.

*"In addition, compared with multifarious models in the fields of deep learning, the structure of GSARNN is relatively lightweight, so its training and calculation efficiency can be quite high. Taking advantage of mighty parallel computing capabilities of GPU units and distributed computing structures to accelerate the*

*training process, the GSARNN model usually converges to the optimal state within 15-20 minutes in our cases. Although the efficiency of Kriging method is better than GSARNN model, under the same condition, it still takes about 10 minutes to fit the functional relationship between the semi-variogram and the distance."*

*"In addition, as the number of sampled points increases, the number of input neurons and output neurons of the GSARNN will also increase, resulting in the expansion of network parameters and the extension of training time inevitably. Therefore, how to maintain a stable and acceptable training time given different sample data volumes is an important problem to be tackled in further researches."*

**Comment 4:** Please express the information of GSDNN unit in Figure 2. Maybe Figure 1 is redundant and it can be merged into Figure 2.

**[Response]**: Thanks for your very instructive advice. The information of GSDNN unit is shown with GSARNN model structure in the same figure (Figure 1 in the revised manuscript) now. Due to the deletion of the original Figure 1, the numbers of subsequent figures and their related text have also been revised accordingly.

[Figure]

*Figure 1. The GSARNN model structure.*

**Comment 5:** In Figure 13, it would be better to make clear that the values in the left column represent depths below the sea surface.

**[Response]**: Thanks for your helpful advice. A description ("section depth") of the

values in the left column in Figure 13 (Figure 12 in the revised manuscript) has been added to avoid readers' confusion.

[Figure]

*Figure 12. Comparisons of interpolated horizontal sections at 100 m depth intervals generated by the four methods.*

**Comment 6:** In Table 1 and Table 4, you'd better change 'Hyperparameters' to 'Hyper-parameters'.

**[Response]**: Thanks for your advice. All the "hyperparameters" in the manuscript

have been revised to "hyper-parameters".

**Comment 7:** When a matrix or a vector is represented by a word or a character, it should be written in bold, such as in Formula 10.

**[Response]**: Thanks for your very helpful advice. All the matrices and vectors in formulas and paragraphs have been revised to bold. In addition, matrices are uniformly represented by upper case letters, and vectors are represented by lower case letters.

**Reviewer 3**

Dear Referee:

Thanks very much for your great support and constructive suggestions with regard to our manuscript. These comments are all valuable and very helpful for revising and improving our paper, as well as the important guiding significance to our researches. We have made our best efforts to improve our paper very carefully following your comments and suggestions. Our point by point response to the comments are given below. We hope the revised manuscript will be acceptable to your requirements. If still there are concerns, we will be happy to take care once we hear from you.

**Comment 1:** In the simulated dataset at section 3.1.1. For component V1, do negative values affect the result? The distance resolution is set as 0.5, why? Why not 1 or 2..?

**[Response]**: Thanks for your comment.

In our simulated case, V1 will not be negative according to our setting. Even if V1 is negative, it will not affect the performance of the GSARNN model. This is because the training process is driven by the loss (MSE in our experiments), and the loss represents the deviation of the interpolated value from the real value, which is equivalent for positive V1 and negative V1.

Setting distance resolution as 0.5, 1 or 2 is equivalent. Different distance resolutions are only different in global magnification, but the relative positions of

sample points and the spatial distribution characteristics of element values are the same.

**Comment 2:** In Section 3.1.2 authors experiment on "1,555 data points in the training set and 173 data points in the validation set". Does the number of data points affect the performance of the algorithm? How?

**[Response]**: Thanks for your very helpful comment. We use the 10-fold cross-validation method for model training, which is a common strategy. The 10-fold cross-validation randomly divides the dataset into 10 equal portions, among which nine portions serve as the training set, and the remaining portion is used as the validation set in turn. This ensures that each sampled point is used as training data and validation data, avoids the impact of data division on results, and fully verifies the performance of interpolation methods. A big fold number will lead to a long training time; a small fold number will cause the training set to be too small to fully learn the data characteristics.

**Comment 3:** For section 3.1.3, what is the settings for OK method?

**[Response]**: Thanks for your very helpful comment. In OK method, we adopt the gaussian model to fit the functional relationship between the semi-variogram and the spatial distance, which turns out to be the optimal variation function model among linear, gaussian, spherical and exponential models. Some explanations have been added in Section 3.1.2, Paragraph 2.

*"Besides, the power parameter of IDW method is 4, and in Kriging method, we adopt the gaussian model to fit the functional relationship between the semi-variogram and the spatial distance, which turns out to be the optimal variation function model among linear, gaussian, spherical and exponential models."*

**Comment 4:** What are limitations and challenges of your algorithm?

**[Response]**: Thanks for your provident comment. In GSARNN model, as the number of sampled points increases, the number of input neurons and output neurons of the

GSARNN will also increase, resulting in the expansion of network parameters and the extension of training time inevitably. This limits the application of the model in scenarios with large amounts of data. How to maintain a stable and acceptable training time given different sample data volumes is a challenge to be tackle in further researches. Some discussions have been added in Section 5 Conclusion.

*"In addition, as the number of sampled points increases, the number of input neurons and output neurons of the GSARNN will also increase, resulting in the expansion of network parameters and the extension of training time inevitably. Therefore, how to maintain a stable and acceptable training time given different sample data volumes is an important problem to be tackled in further researches."*

---

## Author Response (AR2)

**Reply on Editor**

Dear Editor:

We would like to express our great gratitude for you to revise our manuscript. We have made a revision of the manuscript according to the comments and suggestions from you. These comments are all valuable for improving our paper. We sincerely hope that our revisions would meet your requirements. Please don't hesitate to contact us if you have any problems about the response.

**Comment 1:** The authors have addressed the reviewer comments but the literature review is weak. The authors need to revise and improve the literature review and also remove abbreviation in brackets from title.

**[Response]**: Thanks for your very helpful comment. We have reorganized and strengthened the literature review in the introduction section. Besides, the abbreviation and brackets in the title have been removed.

**Comment 2:** The authors need to ensure code and data is provided and enough documentation is provided to execute it.

**[Response]**: Thanks for your reminder. On the basis of the previous version, we have added some doc comments in the code to explain the role or operation mode of each part. The new version of code and data have been re uploaded to the original figshare link in the code and data availability section.
(https://figshare.com/s/8d5e4b6e5b74cc1e0bc1)

**Reply on Reviewer 4**

Dear Referee:

Thanks very much for your great support and constructive suggestions with regard to our manuscript. These comments are very helpful for revising and improving our paper. We have made our best efforts to improve our paper very carefully following

your comments and suggestions. Our point by point response to the comments are given below. We hope the revised manuscript will be acceptable to your requirements. If you still have any concerns, we will be happy to take care once we hear from you.

**Comment 1:** The statement "Unbiasedness means that z*(x0) is the unbiased estimate of z(xi)…" in the newly added explanation of Kriging method (Section 2.1.2) is incorrect. It should read, "z*(x0) is the unbiased estimation of z(x)".

**[Response]**: Thanks for your comment. The statement has been corrected.

**Comment 2:** In the same section, the sentence "Kriging assumes that there is a functional relationship…from the dataset" is either missing a comma or should be split into two sentences.

**[Response]**: Thanks for your very helpful comment. The sentence has been split into two sentences to express the meaning more clearly.

*"Kriging assumes that there is a functional relationship between $r_{ij}$ and $d_{ij}$ (the distance between point $i$ and point $j$). By taking any two sampled points from the dataset, a total of $\frac{n(n-1)}{2}$ $(r, d)$ pairs can be generated."*

**Comment 3:** The matrix computation in formula 13 is Hadamard product, not multiplication, so I am afraid that is mathematically incorrect. Matrix multiplication is not equivalent to element-wise multiplication, unless there is a property of ρ, such as ρ12 + ρ13 + … + ρ1n = 0 written somewhere. Again, the authors failed to address the kij issue raised in Minor Comment 2 by the first reviewer in formula 14.

**[Response]**: Thanks for your very helpful comment. $W$ is the Hadamard product (element-wise product) of matrix $\rho$ and $K$. Formula 13 as well as Formula 10 use the incorrect operating symbol '*'. They have been corrected to '∘'. Besides, the $k_{ij}$ issue in Formula 14 has also been corrected.

$$W = \rho \circ K = \begin{bmatrix} \rho_{11} & \rho_{12} & \cdots & \rho_{1n} \\ \rho_{21} & \rho_{22} & \cdots & \rho_{2n} \\ \vdots & \vdots & \ddots & \vdots \\ \rho_{n1} & \rho_{n2} & \cdots & \rho_{nn} \end{bmatrix} \circ \begin{bmatrix} 0 & 1 & \cdots & 1 \\ 1 & 0 & \cdots & 1 \\ \vdots & \vdots & \ddots & \vdots \\ 1 & 1 & \cdots & 0 \end{bmatrix} = \begin{bmatrix} 0 & \rho_{12} & \cdots & \rho_{1n} \\ \rho_{21} & 0 & \cdots & \rho_{2n} \\ \vdots & \vdots & \ddots & \vdots \\ \rho_{n1} & \rho_{n2} & \cdots & 0 \end{bmatrix} \tag{13}$$

$$\hat{\mathbf{y}} = \begin{bmatrix} 0 & \rho_{12} & \cdots & \rho_{1n} \\ \rho_{21} & 0 & \cdots & \rho_{2n} \\ \vdots & \vdots & \ddots & \vdots \\ \rho_{n1} & \rho_{n2} & \cdots & 0 \end{bmatrix} \begin{bmatrix} y_1 \\ y_2 \\ \vdots \\ y_n \end{bmatrix} = \mathbf{W} * \mathbf{y} \tag{14}$$

**Reply on Reviewer 3**

Dear Referee:

Thanks very much for your affirmation and great support.

---

## Author Response (AR3)

**Reply on Editor**

Dear Editor:

Thanks very much for your kind work and consideration on publication of our paper. On behalf of my co-authors, we would like to express our great gratitude to editor and reviewers.

As requested, there is a minor modification in the latest manuscript. The link in section Code and data availability has been replaced with a DOI link, and the content remains unchanged. If you have any problems about the modification, please don't hesitate to contact us.

*"Simulated data, Argo temperature data and codes used in the study are available at https://doi.org/10.6084/m9.figshare.18739571.v1."*